# How development and survival combine to determine the thermal sensitivity of insects

**Mariana Abarca**[1]*, **Anna L. Parker**[2], **Elise A. Larsen**[3], **James Umbanhowar**[4], **Chandra Earl**[5], **Robert Guralnick**[6], **Joel Kingsolver**[4‡], **Leslie Ries**[3‡]

1 Department of Biological Sciences, Smith College, Northampton, Massachusetts, United States of America, 2 Department of Biology, Wake Forest University, Winston-Salem, North Carolina United States of America, 3 Department of Biology, The University of North Carolina at Chapel Hill, Chapel Hill, North Carolina, United States of America, 4 Department of Biology, Georgetown University, Washington, District of Columbia, United States of America, 5 Department of Natural Sciences, Bernice Pauahi Bishop Museum, Honolulu, Hawaii, United States of America, 6 Florida Museum of Natural History, University of Florida, Gainesville, Florida, United States of America

‡ JK and LR are joint senior authors on this work.
* mabarca@smith.edu

**Data Availability Statement:** Data can be found in Dryad (accession number: DOI: 10.5061/dryad. qjq2bvqk4)

## Abstract

Thermal performance curves (TPCs) depict variation in vital rates in response to temperature and have been an important tool to understand ecological and evolutionary constraints on the thermal sensitivity of ectotherms. TPCs allow for the calculation of indicators of thermal tolerance, such as minimum, optimum, and maximum temperatures that allow for a given metabolic function. However, these indicators are computed using only responses from surviving individuals, which can lead to underestimation of deleterious effects of thermal stress, particularly at high temperatures. Here, we advocate for an integrative framework for assessing thermal sensitivity, which combines both vital rates and survival probabilities, and focuses on the temperature interval that allows for population persistence. Using a collated data set of Lepidopteran development rate and survival measured on the same individuals, we show that development rate is generally limiting at low temperatures, while survival is limiting at high temperatures. We also uncover differences between life stages and across latitudes, with extended survival at lower temperatures in temperate regions. Our combined performance metric demonstrates similar thermal breadth in temperate and tropical individuals, an effect that only emerges from integration of both development and survival trends. We discuss the benefits of using this framework in future predictive and management contexts.

## Introduction

The relationship between performance and temperature is one of the primary factors used in physiological ecology to understand species' evolutionary adaptations to climate and to estimate past, current, and future geographic and temporal distributions. Thermal performance curves (TPCs) depict patterns in vital rates or behaviors in response to temperature and have been an important tool to understand ecological and evolutionary constraints on thermal

**Funding:** This work was funded by Georgetown University and NSF awards: MSB-1702664 and EAGER-1839021 to LR and NSF DEB 1950055 to JGK. The funders had no role in study design, data collection and analysis, decision to publish, or preparation of the manuscript.

**Competing interests:** The authors have declared that no competing interests exist.

sensitivity of ectotherms [1–4]. TPCs have a characteristic unimodal, left-skew shape in which performance gradually increases with temperature up to a maximum, followed by a steep decline as temperature becomes too high for metabolic reactions to occur (Fig 1). This basic shape is consistent for a variety of biological rates across taxa and levels of organization, from individual rates such as growth, development, digestion, and locomotion, to population (e.g. fecundity, population growth) and interaction rates (e.g. predator attack, parasitism; [5]. The consistency of TPC shape allows for the calculation of indicators of ectotherm thermal tolerance (Fig 1) that are subsequently used in comparative studies (e.g. [5]. These metrics include $T_{min}$ and $T_{max}$, the rearing temperatures at which the rate of interest reaches zero, and $T_{opt}$, the rearing temperature that maximizes that rate (Fig 1). Another commonly-computed metric, $T_0$ [6, 7], is estimated by fitting a regression line to the rising linear portion of the curve (black line fitted to black points, Fig 1) and taking its x-intercept.

In turn, a large body of literature has compiled these thermal performance data to make predictions about ectothermic temperature responses along latitudinal [9–13] and altitudinal gradients [14], as well as across different life stages [13, 15]. However, the incorporation of these data into predictive contexts is not straight-forward. Traditionally, TPCs have been created with a performance or fitness proxy, such as locomotion, development rate, or digestion [5] as the response variable. While this strategy is helpful in that it characterizes an ecologically-relevant response for a particular organism, it often underestimates thermal stress at high temperatures: while surviving individuals may appear to perform well due to high vital rates, severe population-level mortality is often observed at high temperatures. Measuring the thermal response of a survivor-biased subsample can be misleading, particularly in comparative studies, where taxa have different evolutionary histories or face different thermal tradeoffs and thus have different adaptations to cope with thermal stress. Similarly, utilizing only survival as a fitness proxy also has its flaws. While organisms may survive a broad range of thermal conditions in a controlled laboratory setting, they often face constraints in nature that are not simulated in the laboratory, such as short growing seasons or predation risk [16]. Surviving, but developing slowly due to thermal limits, may result in decreased fitness in natural settings, limiting the utility of survival-only TPCs.

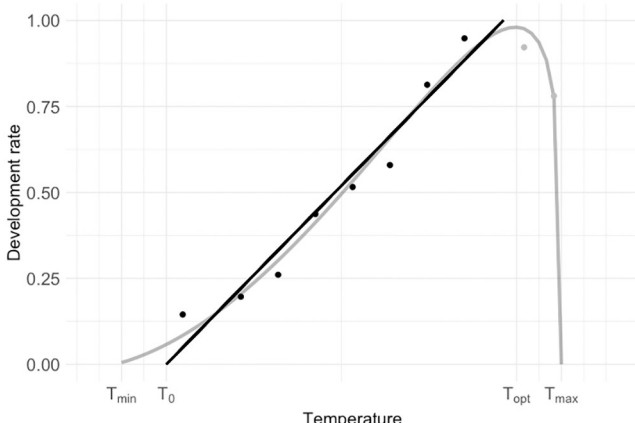

**Fig 1. Thermal performance curve of development rate (grey) and common metrics obtained from it: $T_{min}$, $T_{opt}$, $T_{max}$.** Linear approximation (black) for temperatures below the optimum to obtain the x-intercept, $T_0$. This linear regression applied to development rate data is the basis for developing growing degree-day models, which have been extensively used to develop phenological predictions. Curve generated with data from Butler & Hamilton, 1976 (*Heliothis virescens*, [8].

Neither performance or survival as proxies can identify which underlying process is determining organismal responses at a certain temperature: which process is most limiting to overall performance at low temperatures? At high temperatures? This is particularly important because the shapes of TPCs for different processes can be quite different [4, 17]. For example, development rate TPCs are typically left-skewed with a well-defined intermediate optimum temperature and a rapid decline at temperatures above the optimum (c.f. Fig 1), whereas survival TPCs are typically flat across intermediate temperatures and rapid declines at both low and high temperatures [18]. This basic difference suggests the need for some form of TPC integration. Ideally, a complete performance profile for a given population would be the result of integrating multiple performance curves [17], including responses such as fecundity in addition to survival and development time [19]. However, TPCs on multiple performance metrics are usually limited to extensively studied species such as *Drosophila melanogaster* [20, 21] but see [22], and there are limited data for TPCs for overall fitness (e.g. intrinsic rate of increase) for insects and other ectotherms [5, 23].

Another issue with the extrapolation of individual species' or population's TPCs to general predictions is the choice of parameters to use. Traditionally, thermal minima ($T_{min}$ in Fig 1) and maxima ($T_{max}$ in Fig 1) are used to characterize a species' thermal tolerance [24] and refs therein). While these are useful in that they describe the extremes at which organisms can no longer perform, key ecologically relevant processes stop before an organism ceases to function due to thermal stress [25]. When considering only minimum and maximum temperatures that halt function entirely, it is difficult to predict the effects of stressful but not debilitating temperature exposure. Even when surviving individuals appear to perform well (i.e. fast development) at extreme, but sub-lethal, temperatures, they may have lower fitness due to an unmeasured metric (i.e. reduced fertility due to sterility; [25, 26]. Such temperatures are also less useful when modeling population-level effects because sublethal thermal stress can have severe [27] and cumulative effects that influence population and community dynamics [28].

Here, we advocate for a conceptual framework for assessing thermal sensitivity, which considers both development rates and survival probabilities and focuses on the temperature interval that allows for population persistence [29]. Even though multiple components of fitness could be combined [19], we focus on development and survivorship because these are the two that are most likely to be reported within studies across a broad range of taxa. In particular, we focus on whether interpretation of broad-scale patterns of thermal sensitivity are different when combining survival and development compared to using only one or the other of those variables. Our framework expands on existing literature in two main aspects:

1. We multiply development rate and survival TPCs to create a performance metric **P** (Fig 2), sensitive to differences between these components of fitness and able to identify the factor driving responses at different temperatures.

2. We use 50% performance limits (Fig 2), rather than minima and maxima, to define thermal performance thresholds that exclude the most stressful conditions where individual function is compromised, thus creating more ecologically-relevant limits to interpret.

To illustrate the non-additive effects of incorporating multiple thermal performance measurements when studying broad latitudinal patterns, we collated Lepidopteran thermal tolerance data from empirical research on 59 populations of 40 lepidoptera species. We reconstructed TPCs for development rate (**D**) survival (**S**), and performance (**P**) to estimate lower ($D_l$, $S_l$, $P_l$) and upper ($D_h$, $S_h$, $P_h$) temperature thresholds. These thresholds demarcate the temperature intervals that allow for 50% of the maximum value (i.e. values of .5 or above

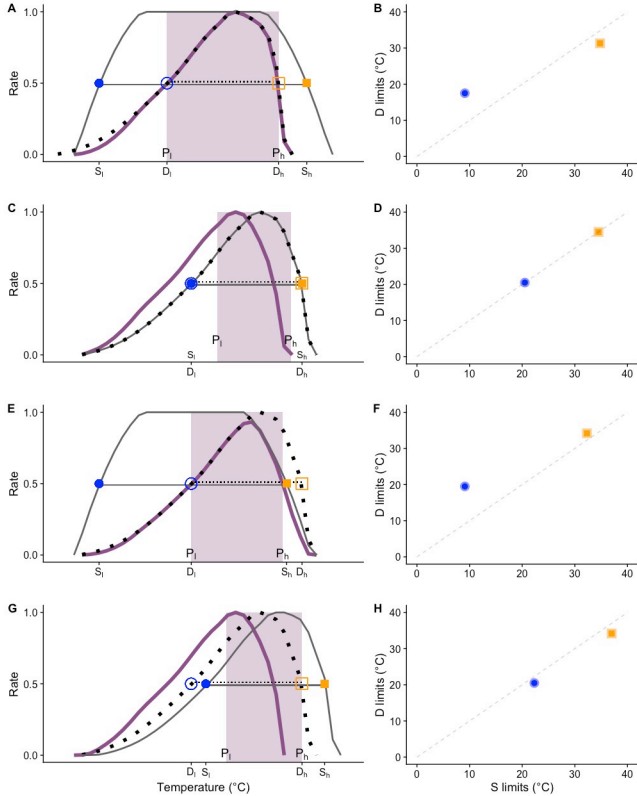

**Fig 2.** Overlap of TPCs (left column) for survival (**S**, solid grey lines) and development (**D**, dotted lines) to determine whether **D** or **S** limit overall performance (**P**, purple solid line) at different temperatures, and the relationships between **S** and **D** limits, both lower (center column, blue circles) and upper (right column, orange squares). Open symbols correspond to **D** limits and solid to **S** limits. Horizontal lines in the left column indicate thermal breadths, and the suitable range (**P_breadth**) is highlighted in purple. Black lines in the right column indicate deviations from the dotted diagonal: vertical deviations indicate development is limiting, while horizontal deviations indicate survival is limiting. **When D limits P at all temperatures**: A) Development limits are contained within survival limits, and B) the low limit would be above the diagonal and the high limit underneath it. **When both D and S limit P**: C) Survival and development limits are the same and D) both points fall along the diagonal. **When S limits P at high temperatures**: E) Development limits are higher than survival limits (**D** curve is shifted to the right) and F) both points fall above the diagonal. **When D limits P at high temperatures**: G) Survival limits are higher than development limits (**S** curve is shifted to the right) and H) both points fall under the diagonal.

on a scale of 0–1) in each of these variables, and can be used to compare populations by characterizing them as cold-tolerant (lower values for $D_l$, $S_l$, $P_l$) or heat-tolerant (higher values of $D_h$, $S_h$, $P_h$). Additionally, the width of these 50% intervals ($D_{breadth}$, $S_{breadth}$) is an indicator of thermal tolerance, and we define the suitable thermal range for a given population as $P_{breadth}$ (Fig 2).

By using this framework, we can assess which component of performance (in this case, development rate or survival) is driving the shape of the TPC over a given temperature range. The different ways in which the individual development rate and survival curves could overlap have distinct consequences. When the interval of temperatures that allows for 50% development rate is located within the interval that maximizes survival, performance is entirely limited by development rate (Fig 2A). When development limits ($D_l$ and $D_h$) are equal to survival limits ($S_l$ and $S_h$), the suitable range is reduced, with $P_l$ being higher than both $D_l$ and $S_l$, and $P_h$ lower than both $D_h$ and $S_h$ (Fig 2C). Another possibility is to have the development curve shifted to the right, in which case both development thresholds are higher than their

corresponding survival thresholds (Fig 2E). By contrast, when the survival curve is shifted to the right, survival thresholds are higher than the corresponding development thresholds (Fig 2G). Each of the different relative patterns illustrated in Fig 2 may predict different evolutionary responses to temperature, and by both comparing and combining relative development and survivorship TPCs, we may see emergent patterns across populations, species, and clades.

Here, we will first explore the relationship between developmental and survivorship metrics to assess variability in the overlapping patterns of development and survival curves across latitude. In addition, we explore how variability in **D**, **S** and **P** limits is explained by phylogenetic history, latitude, and life stage. By comparing latitudinal variation patterns of each of these thermal limits from the same populations, we show that development and survival exhibit non-additive effects, particularly for thermal breadth. Finally, we discuss the utility of 50% performance limits in the context of anthropogenic climate change.

## Methods

### Compiling development and performance data

We compiled a database of published peer-reviewed articles reporting development time and/ or survival of insects from the order Lepidoptera (butterflies and moths) reared at constant temperatures (10.5061/dryad.qjq2bvqk4). We define each experimental unit from which a thermal performance curve can be extracted as a "set": each set consists of 4 to 20 constant temperature treatments (mode = 5) and the corresponding values of mean development time and/or survival. We only included sets containing the most commonly reported life stages: eggs, larvae (hatch to pupation) and pupae (pupation to adult emergence).

When studies reported multiple sets per temperature treatment (i.e. when performance was evaluated in multiple experimental conditions or differentially by sex), we considered sets from the same species, locality, and ontogenetic stage that were reported in the same article to be non-independent. We combined non-independent sets by obtaining the geometric mean at each temperature treatment. We also extracted the latitudinal data from the locality of origin/ collection site for each set and excluded sets that averaged individual responses across collection locations. Some studies did not report the locality of origin of their specimens. These studies typically reported performance of crop pests of local importance in the region of the author's institution. Thus, we assigned them the coordinates of the author's affiliation institution and labeled the locality as "inferred" (S1 Fig in S1 File). Analyses yielded the same qualitative results whether data points from inferred localities were included or not, so we present results including these data and report the percentage of data from "inferred" locality in each particular analysis. After these considerations, our **"full dataset"** consisted of overlapping suites of sets reporting development time (n = 75 species, 173 sets, 24% inferred locations) and/or survival (n = 54 species, 117 sets, 31% inferred locations) data.

In order to make direct comparisons of our novel performance parameter *vs* using development or survival data only, we needed to reduce this dataset further to include only sets that contained both development time and survival data. After this final reduction, our **"analytical dataset"** consisted of 46 sets, from n = 26 species and 17% inferred locations. Not all of the sets, however, contained a "complete curve", or one that had a rise, a peak and a fall, including values of 0.5 or lower at both extremes of the curve. Therefore, we conducted analyses separately for our metrics [5], using all curves with a rise to calculate our low parameters ($D_l$, $S_l$, $P_l$, 37 sets from 21 species and 22% inferred locations), all curves with a fall to calculate our high parameters ($D_h$, $S_h$, $P_h$, 35 sets from 20 species and 11% inferred locations), and only complete curves to calculate our breadth parameters ($D_{breadth}$, $S_{breadth}$, $P_{breadth}$, 26 sets from 15 species and 15% inferred locations). We conducted the analyses below on both the analytical dataset

and the full dataset. After comparison, the trends present in the analytical dataset hold true for the full dataset, thus only those for the analytical dataset are reported below. See S2 File for a description of the full dataset analyses, including model results and figures.

## Calculating thermal performance parameters

The analytical dataset allowed us to quantify thermal performance curves for development rate, survival, and performance (Figs 1 and 2). There is an extensive empirical and theoretical literature on building models (both statistical and mechanistic) to characterize TPCs for development [7, 30]. However, a preferred non-linear model has yet to emerge, likely due to both biological and methodological reasons. There may be slight differences in the real shape of the TPC depending on the taxa and performance metric studied [7], and the observed shapes of the curves depend heavily on the number and identity of temperature treatments (which vary widely among sets in our analytical dataset). This methodological constraint may favor different shapes for spurious reasons or provide too little data to allow more complex models to converge. Because of this, the dominant approach in the insect literature used to characterize the typical TPC has remained simply fitting the linear regression for the rising portion of the curve (solid line in Fig 1; [7]. However, this approach is insufficient for our analyses for two important reasons: first, it does not quantify the shape of the curves across their whole temperature width; and second, the shape of the survival curve (Fig 2A) is not consistent with the typical TPC shape (Fig 1), and thus would require a different modeling approach. Due to these important considerations, we opted instead for simple linear interpolation across the full range of temperatures and computed thermal thresholds based on these interpolations to describe the TPCs. This conservative approach may simplify the curves for which we have fine-scale data; however, this cost is outweighed by the benefit of not biasing against studies with coarser data (fewer temperature treatments, spaced further apart).

For each set, we standardized development rate (the reciprocal of development time) and survivorship so they varied between 0 and 1 (representing the maximum value) and subsequently multiplied them to obtain performance (Performance = Standardized Development Rate × Standardized Survivorship). We then estimated high and low thermal thresholds for survival, development, and performance (Fig 2; $S_l$, $S_h$, $D_l$, $D_h$, $P_l$, $P_h$) using linear interpolation.

## Analyses

To assess overlap patterns of survivorship **S** and development **D** TPCs (Fig 2, left-hand column), we extracted the corresponding thermal limits ($D_l$ and $S_l$; $D_h$ and $S_h$; $D_{breadth}$ and $S_{breadth}$) and used linear regression to evaluate their relationships (Fig 2, middle and right-hand columns). To investigate the effects of ontogenetic stage and latitude on all thermal thresholds and breadths, we used a linear mixed modeling approach. For each response variable (all thermal limits for **S, D,** and **P**), we built a model including ontogenetic stage, latitude, and their interaction as fixed effects, and species as a random effect, using the *lmer* function (package lme4 1.1–23, in R version 4.0.2). To facilitate the interpretation of effect sizes, we fitted the models without an intercept. We used AIC and BIC to select between full (with interaction terms) and additive models. To account for phylogenetic history, we custom built a phylogenetic tree for the 102 species used here. In short, we searched Genbank for 11 full-length, commonly sequenced genomic DNA markers, while accounting for taxonomic issues e.g. synonymy. Five species were not available on Genbank, and in those cases we utilized a congener as a surrogate. Sequences were aligned using mafft v7.294b [31]; Those locus alignments containing 10 or less species were removed, leaving a total of 8 usable loci. We used the

resulting supermatrix and built an unpartitioned maximum likelihood tree using RaxML-NG v0.9.0 [32] under a GTR-G substitution model. See S3 File for full details. With a phylogenetic hypothesis in hand, we fitted the models described above, but including the phylogenetic tree, using the R package *phyr* [33]. Due to the small sample size of our analytical dataset, this phylogenetic model was fitted to our full dataset (found in S2 File). For each response variable, we first fitted a non-phylogenetic model, including species as a random effect (equivalent to the first set of models) and then a phylogenetic model which accounts for phylogenetic autocorrelation. We used partial $R^2$ [34] to compare phylogenetic to non-phylogenetic models. In all cases we obtained very small $R^2_{lik}$ values, indicating a negligible effect of phylogeny [35]; thus, we report throughout results from the regular linear mixed models. Results from the phylogenetic models can be found in S4 File.

## Results

### Thermal performance curve overlap

The lower limit for 50% development ($D_l$) ranged from 13.6 to 28.1 ˚C and was significantly correlated with $S_l$ ($R^2 = 0.4$, $P < 0.0001$, N = 37; Fig 3A), which in turn ranged from 4.1 to 25.9 ˚C. On average, $D_l$ was 5.5 ˚C higher than the corresponding $S_l$, consistent with a right shift of the development curve (Fig 2E and 2F). The slope of this relationship (Fig 3B) indicates that differences between $D_l$ and $S_l$ were greater for those exhibiting relatively low $S_l$ and $D_l$ values (cold-tolerant populations). Variation in higher limits was also consistent with a right shift in the development curve (Fig 2E and 2F), but it exhibited a steeper slope and stronger correlation ($R^2 = 0.78$, $P < 0.0001$, N = 35; Fig 3A), with $D_h$ varying from 26.2 to 39.5 ˚C and $S_h$ from 22.5 to 38.4˚C. On average $D_h$ was 1.2 ˚C higher than the corresponding $S_h$. These results confirm that TPCs are right-shifted for development but not for survival, and that upper thermal limits are generally higher for development than for survival.

Thermal breadth for development ranged from 8 to 18.7˚C (mean ± SD = 12.9 ± 2.19˚C) and was on average 4.7 ˚C narrower than thermal breadth for survival (mean = 16.64 ± 4.7 ˚C, range: 7.7–26.1 ˚C). While $D_{breadth}$ and $S_{breadth}$ were significantly correlated ($R^2 = 0.14$, $P = 0.03$, N = 26; Fig 3C), their relationship exhibited a relatively flat slope (slope estimate = 0.2, Fig 3C), and explains only 3% of the variation.

### Effects of ontogenetic stage and latitude

High thermal performance limits ($D_h$, $S_h$, $P_h$) did not differ much from one another in terms of their relationship to latitude (Fig 4A–4C), as evidenced by the similar latitude coefficients in

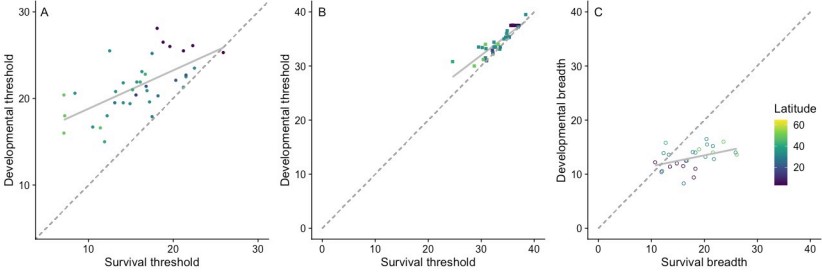

**Fig 3.** Relationship between survival and development for high-temperature limits (A, closed circles), low-temperature limits (B, squares), and thermal breadth (C, open circles). Dashed lines are a reference for exact correspondence; solid lines depict model predictions for each regression. The color gradient represents the latitude of the sets; populations at higher latitudes (yellow) tend to be more cold-tolerant than those at lower latitudes (purple).

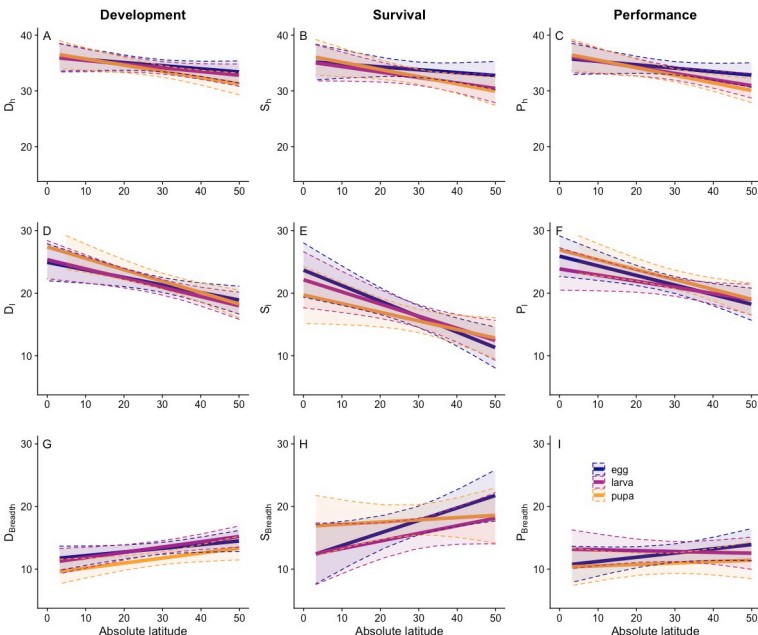

**Fig 4.** Regression lines demonstrating the relationship between development (left column), survival (middle column), and performance (right column) metrics and latitude across life stages (purple: egg; magenta: larva; orange: pupa). Solid lines show model predictions bounded by the dashed lines, which represent 95% confidence interval boundaries.

the linear mixed-effect model results and lack of relationship between latitude and each limit (Table 1A, $P > 0.05$). All of the lower performance limits ($D_l$, $S_l$, $P_l$) varied with latitude (Table 1B, $P < 0.05$), though $S_l$ decreased more dramatically with latitude than development or performance (Fig 4D–4F). This provides confidence that the relationships between our calculated upper and lower thermal limits and latitude are robust to inclusion of any of the three metrics. By contrast, latitudinal patterns in thermal breadth were very different, depending on the variable used to estimate them: $D_{breadth}$ and $S_{breadth}$ both separately increased with latitude (Table 1C, p-values $< 0.05$), but $P_{breadth}$, which is the combination of the previous two, exhibited an almost flat slope (Table 1C, p-value = .520), indicating the same thermal breadth across latitudes (Fig 4G–4I). This demonstrates that, rather than an additive effect of development and survival resulting in an increased response to latitude, the effects of each metric cancel each other out, negating the observed latitudinal pattern.

Thermal thresholds based on development rate were more consistent across life stages than survivorship thresholds. This pattern was observed in analyses of both the analytical (Fig 4, Table 1 in S2 File) and full data sets (Table 2 in S2 File). Specifically, the relationship between $S_{breadth}$ and latitude varied strongly with life stage (Fig 4H, Table 1 in S2 File), with pupal $S_{breadth}$ changing the least across latitudes.

**Table 1. Coefficients with standard errors of absolute latitude in each linear mixed-effects model.** P-values reported in parentheses, with bolded values indicating significance. Complete model outputs, including life stage coefficients and interaction terms (where applicable), can be found in **Table 1 in** S2 File.

|            | Development                    | Survival                       | Performance                    |
|------------|--------------------------------|--------------------------------|--------------------------------|
| A) High    | -0.056 ± 0.042 (0.200)         | -0.050 ± 0.051 (0.336)         | -0.061 ± 0.045 (0.188)         |
| B) Low     | -0.149 ± 0.038 (**0.001**)     | -0.196 ± 0.056 (**0.003**)     | -0.135 ± 0.036 (**0.002**)     |
| C) Breadth | 0.074 ± 0.025 (**0.017**)      | 0.201 ± 0.075 (**0.016**)      | 0.026 ± 0.039 (0.520)          |

## Discussion

By investigating thermal performance in terms of both development and survival, as well as at 50% performance limits rather than thermal extremes, we uncovered key features missing in traditional analyses of thermal performance in ectotherms. We found a consistent pattern of thermal performance curve overlap in which development rate limited performance at low temperatures, while survivorship was the limiting factor at high temperatures. Additionally, when estimating thermal breadth (the favorable temperature range), we detected an unexpected, non-additive effect of survival and development rate. If thermal breadth is estimated using survival or development rate alone, it follows an increasing latitudinal pattern in which temperate species had a larger thermal breadth than tropical species (Fig 4G and 4H). However, when combining both curves, this pattern disappeared, with species exhibiting a thermal breadth of ~12 ˚C regardless of latitude (Fig 4I). Below, we discuss the implications of our findings in the context of adaptation to global change and call for an integration of thermal performance curves to better understand ectotherm thermal sensitivities.

### General patterns of curve overlap

We found a consistent pattern of thermal performance curve overlap in which development thresholds for both high and low limits are higher than survival thresholds. Thus, our results (Fig 3A and 3B) most closely resemble the hypothetical relationship demonstrated in Fig 2E and 2F. This pattern, in which each $S_l$ has a lower value than the corresponding $D_l$, indicates that development rate is limiting at low temperatures, while survival is limiting at high temperatures (Fig 2E and 2F). While this general pattern was overall consistent across latitudes, we found that the difference between $D_l$ and $S_l$ limits was greater at high latitudes (Fig 3A), with temperate populations exhibiting lower survival limits than tropical populations, but all populations exhibiting similar development rate limits regardless of origin. Variation patterns in survival reflect the variety of physiological adaptations that allow temperate species to withstand cold stress at temperatures significantly lower than those that permit metabolism [36]– some at temperatures as low as -40 ˚C [37]. These adaptations include both immediate responses to cold stress in metabolically active life stages (e.g. cold stupor or chill coma responses [38, 39] and those of metabolically inactive life stages (e.g. diapause or hardiness to sub-freezing temperatures [40]. On the other hand, adaptations to heat stress involve the production of heat shock proteins, but are very metabolically expensive and only allow for survival at temperatures a few degrees above the metabolic optimum [41]. We found that the overlap and overall shape of **S** and **P** curves reflect this pattern, in which the range of survival optima is predominantly lower than the singular development optimum, extending the survival curve farther towards the low-temperature end of the spectrum (Fig 2E). Consistently, the interval allowing for 50% survival, $S_{breadth}$, was larger than the corresponding threshold for development, $D_{breadth}$, especially at high latitudes (Fig 3C). At high temperatures, however, the close values of $S_h$ and $D_h$ leave less room for selection to act to improve overall performance. By calculating thermal limits that allow for 50% performance, we can identify temperature ranges where selection would be likely to occur as opposed to ranges where survivorship is too low to allow for population persistence. These limits can be used to identify geographic areas of interest for conservation purposes, where populations may be rescued via selection.

### Differences across life stages and latitudes

Our analysis of both development and survival curves indicates differences in thermal parameters across life stages that were not detectable from the development curves alone (Fig 4) that warrant further analyses. Specifically, across life stages, $S_l$ showed greater variation at low

latitudes (Fig 4E) than $D_l$ (Fig 4D), and $S_h$ showed greater variation at high latitudes (Fig 4B) than $D_h$ (Fig 4A). This pattern is consistent with the different selective pressures that would be expected across latitudinal gradients, with heat tolerance selected for in tropical populations and cold tolerance in temperate populations. While activation temperature for metabolic reactions is fairly conserved across taxa [5], functional traits and ecological roles can trigger deviations from this pattern; for example, Dell et al. [5] showed that prey had lower activation energy than predators, reflecting the asymmetric nature of predator-prey interactions. In holometabolous insects, different life stages have contrasting ecological roles and can occur at different times of the year: thus, they face different selective pressures. Consistently, we found that variation in thermal limits occurred not only across latitudinal gradients, but also across life stages (S2 File). This is supported by work in other ectothermic systems [13, 42, 43] that demonstrates changing thermal parameters depending on life stage. Kingsolver and Buckley [13] found changes in $T_0$ based on ontogenetic stage and latitude to be highly variable between species as well.

The most striking difference observed in a performance metric is the relationship between $P_{breadth}$ and latitude (Fig 4I). As expected, $D_{breadth}$ (Fig 4G) and $S_{breadth}$ (Fig 4H) increased with increasing latitude, as species experience a wider range of temperature conditions in temperate climates and must be prepared to withstand them [24, 44]. However, when combined, those relationships disappear entirely in larvae and pupae, and lessen in magnitude for eggs (flat slopes of regression lines in Fig 4I). This suggests that the development and survival curves do not overlap in a manner that increases performance (Fig 2E); rather, the curves are offset from one another along the temperature x-axis such that their combination does not reflect the full breadth of either curve. In other words, at high latitudes, the development and survival optimum temperatures are separated along the temperature x-axis, negating the positive effects of increased thermal breadth in each metric alone. This is consistent with recent studies [29, 45] but differs from the conclusions of previous research that utilize traditional metrics such as $T_{min}$ or $T_{max}$ as an estimate of thermal tolerance [2, 46, 47]. Such studies assert that broader thermal tolerance at high latitudes is expected due to lower $T_{min}$ values and explained by climatic conditions; indeed, it is the pattern we expected to find with our analyses. From a more holistic performance viewpoint, however, this assumption does not hold: a testament to the necessity of including survival data in thermal tolerance calculations. It is important to note, however, that our analytical dataset includes mainly sets from intermediate latitudes (see S1 Fig in S1 File); thus, points from very low or high latitudes will have a large influence on the reported relationship between latitude and our metrics. We hope that future research from these regions will solidify our findings.

## Future directions and applications

As the planet continues to warm, organisms will be affected holistically by changing mean and extreme temperatures [48]. Understanding how increasing temperatures will affect each facet of an organism's performance is critical to making predictions about its success. Moreover, measuring metrics that are relevant on a population-level scale are crucial to extrapolating individual-level effects onto whole species or geographic ranges. Due to their integrative, holistic nature, the performance metrics we propose are more comprehensive than previously-utilized ones that focus on survival or development alone. By calculating values that represent a temperature range at which functions are sustainable, not when either are optimized or completely halted, we can make more informed predictions about population persistence or success. These thresholds can also be applied to identify areas where selection still has the capacity to increase thermal tolerance and/or fitness; we believe understanding the

implications of the selection landscape on insect thermal tolerance will be a crucial area of future study as climate change progresses.

Our results regarding $P_{breadth}$ are critical when considering insect responses to climate change. A general expectation that comes from looking at critical thermal limits is that temperate species should be more able to survive warming temperatures [2, 24], due to having a broader thermal range and currently inhabiting a lower region of that thermal range [49]–although phenological or life history patterns may reduce that survival ability (see [50]. Broadly, this should result in temperate species having enough time for either a) selection to act and their thermal tolerance to increase, or b) populations to disperse towards the poles or upward in elevation. However, the observed overlap pattern between development and survival curves at high latitudes offer a key counterpoint. Since survival is limiting at high temperatures, and the expansion of $S_{breadth}$ at high latitudes is occurring on the cool end of the range (not the warm end), we would not necessarily expect that temperate populations would be better able to withstand higher temperatures [24]. Also, we did not see a broader thermal tolerance at high latitudes, as $P_{breadth}$ did not increase with latitude (Fig 4I). Combined, this suggests we cannot assume that temperate species exhibit increased persistence under warming global conditions. However, it is important to highlight that we do not consider tolerance to thermal conditions above our $P_h$ metric (between $P_h$ and $T_{max}$), which may have severe consequences on organismal persistence [51, 52] and further modify our predictions.

The framework we propose can be implemented by modelers and empiricists alike to quantify thermal tolerance, especially in insect systems. As the study of thermal limits increases in popularity due to climate change, further integration of thermal performance curves will be possible. We advocate for a broader experimental approach that includes developmental and survival metrics when making TPCs, to aid in population-level predictive efforts. As we attempt to conserve at-risk species and maintain ecosystem services in the face of vast, heterogeneous insect declines world-wide [53, 54], integrative approaches that further improve our predictive toolkit are imperative.

## Supporting information

**S1 File. List of species and location of populations included in analyses.**
(DOCX)

**S2 File. Model results from the "analytical" and "full" data sets.**
(DOCX)

**S3 File. Phylogenetic analyses.**
(DOCX)

**S4 File. Models with phylogenetic correction.**
(DOCX)

## Acknowledgments

We are thankful to the team that participated in data gathering: Andra Doherty, Sophie Lokwood, Madeline Lee, Declan Mirabella, Kevin Czachura, Jonas Goldman, and Erin Leeds.

## Author Contributions

**Conceptualization:** Mariana Abarca, Elise A. Larsen, Joel Kingsolver, Leslie Ries.

**Data curation:** Mariana Abarca, Anna L. Parker, Elise A. Larsen.

**Formal analysis:** Mariana Abarca, Chandra Earl.

**Funding acquisition:** Joel Kingsolver, Leslie Ries.

**Methodology:** Mariana Abarca, James Umbanhowar, Robert Guralnick, Joel Kingsolver, Leslie Ries.

**Project administration:** Mariana Abarca.

**Resources:** Leslie Ries.

**Supervision:** Mariana Abarca.

**Visualization:** Mariana Abarca, Anna L. Parker.

**Writing – original draft:** Mariana Abarca, Anna L. Parker.

**Writing – review & editing:** Mariana Abarca, Anna L. Parker, Elise A. Larsen, James Umbanhowar, Chandra Earl, Robert Guralnick, Joel Kingsolver, Leslie Ries.

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
