## [Decision Letter · Decision Letter 0]

5 Jul 2023

PONE-D-23-02122How development and survival combine to determine the thermal sensitivity of insectsPLOS ONE

Dear Dr. Abarca,

Thank you for submitting your manuscript to PLOS ONE. After careful consideration, we feel that it has merit but does not fully meet PLOS ONE’s publication criteria as it currently stands. Therefore, we invite you to submit a revised version of the manuscript that addresses the points raised during the review process.

We look forward to receiving your revised manuscript.

Kind regards,

Daniel de Paiva Silva, Ph.D.

Academic Editor

PLOS ONE

Journal Requirements:

This work was funded by Georgetown University and NSF awards:  MSB-1702664 and  EAGER-1839021 to LR and NSF DEB 1950055 to JGK.

Reviewers' comments:

Reviewer's Responses to Questions

**Comments to the Author**

1. Is the manuscript technically sound, and do the data support the conclusions?

Reviewer #1: Yes

Reviewer #2: Yes

2. Has the statistical analysis been performed appropriately and rigorously? 

Reviewer #1: I Don't Know

Reviewer #2: Yes

3. Have the authors made all data underlying the findings in their manuscript fully available?

Reviewer #1: Yes

Reviewer #2: Yes

4. Is the manuscript presented in an intelligible fashion and written in standard English?

Reviewer #1: Yes

Reviewer #2: Yes

5. Review Comments to the Author

Reviewer #1: Dear authors,

I have had the opportunity to review your manuscript, and I would like to express my appreciation for the valuable contribution you have made to the field of thermal biology. The study is well-conducted and the manuscript is well-written, providing important insights into the thermal sensitivity of insects.I have thoroughly reviewed your manuscript and found it to be exceptionally well-crafted with no significant concerns or shortcomings. This level of quality and attention to detail is truly commendable.

I would like to offer a suggestion for improvement regarding the detailing of the construction of the phylogenetic tree. Please provide the reader with more details on the phylogenetic methods employed. This will provide more clarity on the phylogeny implications of your results. Once this concern is addressed, I am confident that your work will make a significant contribution.

Thank you for the opportunity to review your manuscript.

Reviewer #2: Overview

In this manuscript, the authors use models to overlap the development and survival performance curves of an array of Lepidopteran species in order to determine which affects their persistence in an area the most, as well as what effects sub-lethal impacts might have on their performance. This was done across different life stages and latitudes.

This is a very well and clearly written manuscript that I believe will be accessible by a reader who is not an expert in the field. The majority of my comments below are purely just to add a little clarity or higher quality reader experience here and there rather than commenting on issues with the study itself or the writing of the manuscript. I have selected ‘minor revision’ for the paper only because there are a very small number of changes that I recommend below under the ‘specific comments’ heading.

The question of incorporating multiple different physiological measurements into one single profile for an animal while simultaneously considering sub-lethal effects is something that I have pondered on many an occasion, although I had not combined the two thoughts (single profile and sub-lethal effects) into one yet. Because this manuscript answers questions that I have thought about myself, as well as had discussions with many other thermal physiologists about, I believe it is an important paper with high relevance to its field. Thank you to the authors for answering these questions that have plagued me but that I have not had the chance to focus my full attention on answering yet! Furthermore, this paper directly responds to the growing trend of moving away from single-limit investigations and considering animals more holistically, particularly in the context of climate change. I believe that it will become a staple reading for all researchers working in the field of thermal ecophysiology.

General comments

Abstract

This is a good abstract that encompasses all of the sections well. I have no suggested changes here.

Introduction

I find the arguments presented here to be valid and sound. They are backed up and responsive to other published literature. A sufficient and detailed background is provided to support the approach taken, which is cleverly done and fully captures what the authors state they aim to achieve. I put some thought into whether the paragraphs from line 130 onwards belonged in the introduction or the discussion rather. However, I ultimately decided that they fit better in the introduction where they currently are to give background on how the models were due to work. Finally, aims for the paper are given, which aim to test the developed models. I have no general suggestions for changes in this section.

Methods

I believe that this study would be reproducible from the methods as presented here. That said, I wonder if an infographic or flowchart of some kind would help the reader fully grasp the process described in lines 180-205? This is not vital for the publication of this manuscript, I think it would just add a little extra clarity if such a figure is possible to make. The steps described in this section in order to prepare the data for analysis make sense to me and are presented in a logical way, a visual representation would just be a nice addition. The supplementary information is appreciated to back the study and its conclusions up.

The arguments presented in the section titled ‘Calculating thermal performance parameters’ that support the methodological decisions made are logical and valid. I think it is valuable to show these background workings that went into the development of the ideas in the manuscript to validate the final steps. Furthermore, the statistical models used make sense to me.

Results

The results are well-described and logical to follow. The general trends described here align with what I have found in my own studies on individual species and my understanding of thermal tolerance, which I wanted to highlight to demonstrate that they make sense. I have no general suggested changes here.

Discussion

This is a good discussion where the claims are fully supported by the data and analyses from the results section. The conclusion that development limits performance at low temperatures, while survival limits it at high temperatures, is logical. I am glad to see it so clearly stated here with support from the results. The finding that the thermal breadth for all species across latitudes was very similar is extremely interesting. Perhaps this is the first support for a ‘12deg C performance law’ for insects?!?

The first paragraph in the discussion is very good at getting the message of this manuscript across in a clear and punchy way. The rest of the discussion thereafter makes the findings very clear and digestible. Valid explanations that are supported by sufficient and relevant literature are also presented for each observed trend. It is fantastic to see this discussion extended beyond thermal tolerance to selection for population persistence too.

The final section on future directions and applications is very good. I completely agree with all points raised here. This expansion of the study’s findings to understanding how species might respond to climate change is brilliant and should make the reader challenge what they have been thinking thus far, as it did for me.

Specific comments

Title

Short title: I think this could be shortened and made punchier

Introduction

Line 85: Replace “lab” with “laboratory”

Methods

Line 182: A comment has been left in the text to “insert data repository info”, which needs to be done

Line 124: I would remove the semi-colon and split this sentence into two as follows “…full dataset. After comparison…”

Line 239-241: Would it not look better and be clearer if a formula was provided here? E.g. Performance = Standardized Development Rate x Standardized Survivorship. This will just help the reader find this easier if they are following the method and looking for the formula.

Results

Figure 3: Perhaps it is just the version of PDF that I downloaded, but the found the quality of all figures to be quite poor. This was particularly important for Figure 3, where I struggled to tell circles and squares apart as a result. I hope that the figures submitted for final publication are of a better quality.

Table 1 and Figure 4: This table and figure are misplaced. They are first referred to in the paragraph starting on line 292, so I think they should be moved to line 306.

6. PLOS authors have the option to publish the peer review history of their article (what does this mean?). If published, this will include your full peer review and any attached files.

Reviewer #1: No

Reviewer #2: **Yes: **Candice Ann Owen

---

## [Author Response · Author response to Decision Letter 0]

3 Aug 2023

Dear Dr. de Paiva Silva, 

We would like to thank you, Dr. Owen and reviewer #1 for your thoughtful comments and suggestions, which allowed us to prepare an improved version of our manuscript “PONE-D-23-02122. How development and survival combine to determine the thermal sensitivity of insects”. 

 In this new version we include a description of how we built the phylogenetic tree used in our analyses, we edited the text to comply with PLOS ONE formatting guidelines, we included the accession number Dryad assigned to the data presented here and we updated the information about our funding sources (see cover letter). 

We thank both reviewers for their positive feedback and constructive comments that allowed us to improve our manuscript. Please find below a list of the reviewers’ suggestions and descriptions on how we addressed each of them.

We would like to update our financial disclosures to state that: 

List of responses to reviewer comments:

Reviewer #1. 

1- “I would like to offer a suggestion for improvement regarding the detailing of the construction of the phylogenetic tree. Please provide the reader with more details on the phylogenetic methods employed. This will provide more clarity on the phylogeny implications of your results. Once this concern is addressed, I am confident that your work will make a significant contribution.”

We included the following statement in the revised manuscript explaining our phylogenetic methods (lines 311-318): 

“To account for phylogenetic history, we custom built a phylogenetic tree for the 102 species used here. In short, we searched Genbank for 11 full-length, commonly sequenced genomic DNA markers, while accounting for taxonomic issues e.g. synonymy. Five species were not available on Genbank, and in those cases we utilized a congener as a surrogate. Sequences were aligned using mafft v7.294b [31]; Those locus alignments containing 10 or less species were removed, leaving a total of 8 usable loci. We used the resulting supermatrix and built an unpartitioned maximum likelihood tree using RaxML-NG v0.9.0 [32] under a GTR-G substitution model. See supporting information S3 for full details. With a phylogenetic hypothesis in hand, we fitted the models described above, but including the phylogenetic tree, using the R package phyr [33]”

We also updated Supplement 3, which describes in more depth the process outlined above, including a table of the DNA markers we used and a list of substitutions due to lack of DNA information.

Reviewer #2: 

Methods

1. I believe that this study would be reproducible from the methods as presented here. That said, I wonder if an infographic or flowchart of some kind would help the reader fully grasp the process described in lines 180-205? This is not vital for the publication of this manuscript, I think it would just add a little extra clarity if such a figure is possible to make. 

We sketched an infographic but did not consider that it would improve the clarity of the methods described in lines (now 196 to 217) so we decided to keep the original version of this section. 

2. Short title could be shortened and punchier.

We changed the short title to “Insect thermal sensitivity”

3. Line 85: Replace “lab” with “laboratory”

We changed “lab” to “laboratory” as suggested (now line 116)

4. Line 182: A comment has been left in the text to “insert data repository info”, which needs to be done

We included Dryad information (line 223):- DOI: 10.5061/dryad.qjq2bvqk4

5. Line 124: I would remove the semi-colon and split this sentence into two as follows “…full dataset. After comparison…”

We think this refers to line 214 (now 235). We removed the semi-colon as suggested.

6. Line 239-241: Would it not look better and be clearer if a formula was provided here? E.g. Performance = Standardized Development Rate x Standardized Survivorship. This will just help the reader find this easier if they are following the method and looking for the formula.

We included the formula as suggested (lines 292-293): 

“For each set, we standardized development rate (the reciprocal of development time) and survivorship so they varied between 0 and 1 (representing the maximum value) and subsequently multiplied them to obtain performance (Performance = Standardized Development Rate × Standardized Survivorship).”

Results

7. Figure 3: Perhaps it is just the version of PDF that I downloaded, but the found the quality of all figures to be quite poor. This was particularly important for Figure 3, where I struggled to tell circles and squares apart as a result. I hope that the figures submitted for final publication are of a better quality.

We updated all of our figures using PACE to make sure they meet PLOS format requirements.

8. Table 1 and Figure 4: This table and figure are misplaced. They are first referred to in the paragraph starting on line 292, so I think they should be moved to line 306.

We moved Table 1 and Figure 4 so they now appear at the end of the paragraph where they are first mentioned.

---

## [Decision Letter · Decision Letter 1]

29 Aug 2023

How development and survival combine to determine the thermal sensitivity of insects

PONE-D-23-02122R1

Dear Dr. Abarca,

We’re pleased to inform you that your manuscript has been judged scientifically suitable for publication and will be formally accepted for publication once it meets all outstanding technical requirements.

Kind regards,

Daniel de Paiva Silva, Ph.D.

Academic Editor

PLOS ONE

Additional Editor Comments (optional):

Dr. Abarca,

Congratulations on your hard work on improving your manuscript! I am pleased to inform that your manuscript has been accepted for publication in PLoS One!

Best regards,

Daniel Silva.

Reviewers' comments:

Reviewer's Responses to Questions

**Comments to the Author**

1. If the authors have adequately addressed your comments raised in a previous round of review and you feel that this manuscript is now acceptable for publication, you may indicate that here to bypass the “Comments to the Author” section, enter your conflict of interest statement in the “Confidential to Editor” section, and submit your "Accept" recommendation.

Reviewer #1: All comments have been addressed

Reviewer #2: All comments have been addressed

2. Is the manuscript technically sound, and do the data support the conclusions?

Reviewer #1: Yes

Reviewer #2: (No Response)

3. Has the statistical analysis been performed appropriately and rigorously? 

Reviewer #1: Yes

Reviewer #2: (No Response)

4. Have the authors made all data underlying the findings in their manuscript fully available?

Reviewer #1: Yes

Reviewer #2: (No Response)

5. Is the manuscript presented in an intelligible fashion and written in standard English?

Reviewer #1: Yes

Reviewer #2: (No Response)

6. Review Comments to the Author

Reviewer #1: Dear authors

I am writing to inform you that I have carefully reviewed the revised version of your manuscript titled "How Development and Survival Combine to Determine the Thermal Sensitivity of Insects." I am pleased to convey that all of my previous concerns have been thoroughly addressed, and I would like to commend you and your co-authors for the meticulous effort you have put into incorporating my suggested additions.

The manuscript has evolved into an even more captivating read, offering valuable insights into the intricate interplay between development, survival, and thermal sensitivity in insects. Your study promises to make a significant advancement in our understanding of this subject matter, and I believe it will undoubtedly stimulate further research in the field.

Thanks for the opportunity.

Reviewer #2: (No Response)

7. PLOS authors have the option to publish the peer review history of their article (what does this mean?). If published, this will include your full peer review and any attached files.

Reviewer #1: **Yes: **Vinicius Marques Lopez

Reviewer #2: **Yes: **Candice Ann Owen

---

## [Editor Report · Acceptance letter]

19 Sep 2023

PONE-D-23-02122R1 

How development and survival combine to determine the thermal sensitivity of insects 

Dear Dr. Abarca:

I'm pleased to inform you that your manuscript has been deemed suitable for publication in PLOS ONE. Congratulations! Your manuscript is now with our production department. 

Kind regards, 

on behalf of

Dr. Daniel de Paiva Silva 

Academic Editor

PLOS ONE